# Deciphering the RRM-RNA recognition code: A computational analysis

**Joel Roca-Martínez**[1,2], **Hrishikesh Dhondge**[3], **Michael Sattler**[4,5], **Wim F. Vranken**[1,2¤]*

**1** Interuniversity Institute of Bioinformatics in Brussels, VUB/ULB, Brussels, Belgium, **2** Structural biology Brussels, Vrije Universiteit Brussel, Brussels, Belgium, **3** Université de Lorraine, CNRS, Inria, LORIA, Nancy, France, **4** Institute of Structural Biology, Molecular Targets and Therapeutics Center, Helmholtz Munich, Neuherberg, Germany, **5** Bavarian NMR Center, Department of Bioscience, School of Natural Sciences, Technical University of Munich, Garching, Germany

¤ Current address: Interuniversity Institute of Bioinformatics in Brussels, VUB/ULB, Brussels, Belgium.
* wim.vranken@vub.be

**Data Availability Statement:** All the datasets and code required to run RRMScorer is publicly available at https://bitbucket.org/bio2byte/rrmscorer/src/master/.

## Abstract

RNA recognition motifs (RRM) are the most prevalent class of RNA binding domains in eucaryotes. Their RNA binding preferences have been investigated for almost two decades, and even though some RRM domains are now very well described, their RNA recognition code has remained elusive. An increasing number of experimental structures of RRM-RNA complexes has become available in recent years. Here, we perform an in-depth computational analysis to derive an RNA recognition code for canonical RRMs. We present and validate a computational scoring method to estimate the binding between an RRM and a single stranded RNA, based on structural data from a carefully curated multiple sequence alignment, which can predict RRM binding RNA sequence motifs based on the RRM protein sequence. Given the importance and prevalence of RRMs in humans and other species, this tool could help design RNA binding motifs with uses in medical or synthetic biology applications, leading towards the *de novo* design of RRMs with specific RNA recognition.

## Author summary

The interactions between proteins and RNAs are key to many different biological processes and crucial for the proper functioning of the cells. The RNA recognition motif is the most prevalent protein carrying such functions in eucaryotes, so understanding how this protein motif interacts with different RNA sequences is of immediate relevance. However, a general recognition code between this motif and the RNA is not known yet. We have performed a computational analysis to understand how the recognition process works, identify the main binding mode between the protein and the RNA and which are the most relevant amino acids involved in the recognition. This analysis allowed us to build a predictor that estimates the binding between an RNA recognition motif and any RNA sequence. We have named this method RRMScorer and it only needs the sequence of both the protein and the RNA to predict the results. We hope that this new tool will be useful to identify new potential RNA targets or to design new protein mutants to modify their RNA binding capabilities.

**Funding:** J.R-M., H.D., M.S., and W.V.; Marie Skłodowska-Curie Innovative Training Network (MSCA-ITN) RNAct supported by European Union's Horizon 2020 research and innovation programme under grant agreement No 813239. (https://ec.europa.eu/info/research-and-innovation/funding/funding-opportunities/funding-programmes-and-open-calls/horizon-2020_en). The funders had no role in study design, data collection and analysis, decision to publish, or preparation of the manuscript.

**Competing interests:** The authors have declared that no competing interests exist.

## Introduction

The RNA recognition motif (RRM) is a well-studied RNA-binding domain that is prevalent throughout organisms, but especially so in eucaryotes, where it plays crucial roles in many aspects of post-transcriptional gene regulation [1]. A single RRM domain is approximately 90 residues long, with a very conserved topology of two α-helices packed on an antiparallel β-sheet. The four β-strands and the loops connecting the secondary structure elements (Loop1 connecting β1- α1, loop 3 connecting β2-β3 and loop 5 connecting α2-β4) often serve as the main RNA binding interface [2] (Fig 1A). Even though the core 3D fold of the RRM is very conserved, its amino acid sequence has exhaustively evolved to specifically bind different RNA sequences [3], thereby enabling RRMs to regulate a wide range of biological functions [2]. Most commonly, the RRM binds single stranded RNA (ssRNA), and in some cases single stranded DNA [4,5] or structured RNA motifs [6]. Examples are available where protein-protein interactions between an RRM and another RRM, or a non-RRM protein, can modulate the RRM binding capabilities [7–9] and the RRM fold has also evolved to mediate protein-protein interactions with limited or no RNA binding capability, e.g. U2AF Homology Motif (UHM) domains that recognize peptidic UHM Ligand Motifs (ULMs) [10,11].

Many efforts have been made to understand the RNA recognition mechanism, and despite the identification of some RNA consensus sequences for several RRM domains [12–19], many RRMs have no consensus sequence identified yet. The variable RNA binding modes and variations in RRM subfamilies [1] complicate the identification of a general code for RRM-RNA recognition, which has remained a challenge for many years [20]. The increasing number of structures of RRM-RNA complexes now available allows a more detailed and general analysis of the residue-nucleotide preferences of RRMs. RNA binding preference predictions for RNA binding proteins (RBP) have already been a research focus during the last decade, where deep learning methods are taking advantage of the huge amount of information available for certain RBP families [21,22]. One of the main limitations of these methods is that often only the RNA information is considered [18,23], which is unfeasible for RRMs where a few amino acid changes on the RNA binding interface can completely change RNA specificity [24,25]. Knowledge-based potentials have also been widely used to study protein/nucleic acid interactions [26,27], with some specific applications on protein-RNA recognition [28]. One of the main limitations of this method is that they rely on docking models and a detailed calculation of all the atomic interactions, and therefore have a strong dependence on the precise structural data that the potential was based on [29]. The recent release of RoseTTAFoldNA [30] also promises a huge advance in the field, providing high accuracy models with atomic resolution for protein/nucleic-acid complexes, which is extremely useful for proteins where it is clear which RNA the protein binds, but that is not always the case. Therefore, a fast and interpretable method that works at the sequence level and that is applicable to genome scale studies to identify possible interactions, or that can be used in computational screening in protein design, is not available.

Here we introduce RRMScorer, a scoring method that overcomes these limitations by carefully combining information on protein and RNA, structure and sequence data, whilst minimising bias in both areas. A comprehensive analysis of these data reveals RRM-RNA interaction preferences, which are used to generate a score that informs how likely a particular RRM-RNA interaction is. We focus on the canonical binding mode characterized by the binding of the RNA to the surface of the central β-sheet of the RRM fold, (Fig 1A), which comprises aromatic residues in the β3 and β1 strands that mediate non-sequence-specific stacking interactions with RNA bases. These residues are part of the highly conserved ribonucleoprotein domain 1 (RNP1) and RNP2, in the β3 and β1 strands, respectively, and are a hallmark of the

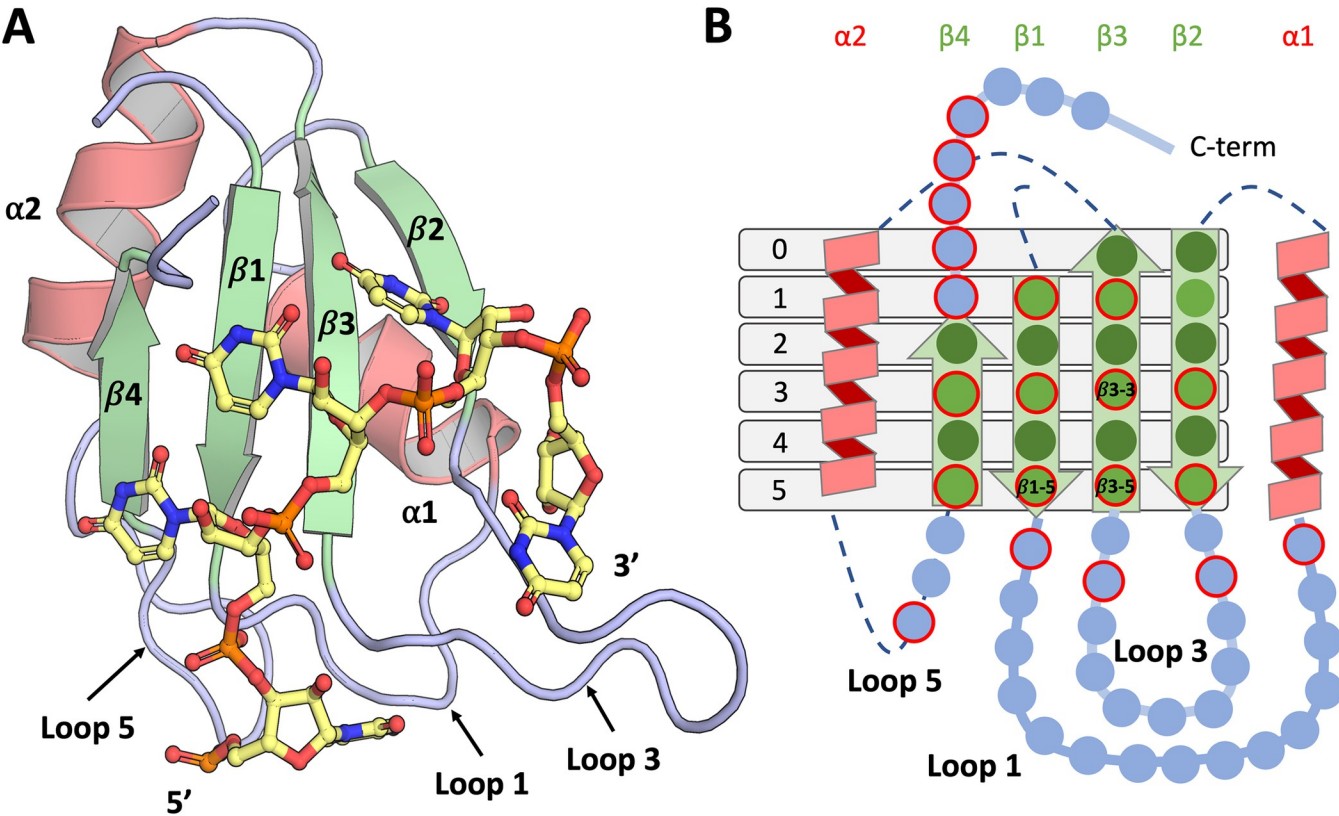

**Fig 1.** A) Cartoon representation of the Sex-lethal RRM1 protein in complex with polyU (PDB Id. 1b7f). The main secondary structure elements are labelled and coloured in pale green (β-strands), dark salmon (α-helices) and light blue (loops and terminal regions). B) Schematic representation of the RRM depicting the main positions of the canonical RNA binding interface with single circles and highlighting in red the most significant ones in terms of RNA interactions prevalence. The same colour code is used and the light green and dark green circles in the β-strands correspond to the exposed and buried residue sidechains, respectively.

canonical RRM fold. Even within this canonical binding mode, a wide range of different RNA sequences can be recognized. Indeed, some of the well-described human RRMs such as HuR, U1A and PTB (UniProt Ids. Q15717, P09012, P26599) recognize very different RNA sequences [19] whilst sharing the canonical binding mode.

This meta-analysis of available RRM-RNA information brings us closer to understanding how this versatile motif works. RRMScorer provides a novel means to decipher a general recognition code for canonical RRMs by using a completely different approach from other RBP predictors, where complex deep-learning networks are trained using high-throughput RNA binding data to determine the consensus for different RRM domains [21,22]. Instead, our method relies on a carefully curated alignment for all the structural information available, that is then translated into a single score that estimates how likely it is that any residue in a specific position of the RRM interacts with any nucleotide of the RNA. One of the key points of RRMScorer is therefore its interpretability, enabling tracking of the residue-nucleotide contacts that lead to good or bad overall scores for an RRM-RNA complex. This approach brings us closer to successfully designing novel RRMs that are specific for different ssRNA targets. Due to the wide range of processes RRMs are involved in, designing such RRMs would be relevant in both therapeutic and synthetic biology fields, for example through creating novel means for post-transcriptional regulation through RRMs, as well as for discovery of *in vivo* RNA targets of RRMs with as yet unknown function and interactions.

## Materials and methods

### Source data

To gather the RRM data available we started from the Pfam database [31], where 19 families containing RRMs were identified (S1 Table) and further validated through visual inspection of representative structures for each family. All the available entries on the Protein Data Bank belonging to those families were retrieved and split into the separate RRM domains. The structures can be found in proteins containing a single RRM domain to larger proteins containing up to four RRM domains. 1259 RRM domain structures were so identified in the complete RRM dataset (S1 Dataset), from which the domains in complex with RNA were extracted. 271 entries remained after removing the complex structures where less than three nucleotides are recognized (Fig 2). The complete RRM dataset and the RRM-RNA complex dataset (S2 Dataset) are available from the supplementary information folder in the Bitbucket repository. All entries are identified by their UniProt code, RRM number, PDB Id. and chain, starting and ending positions of the RRM by PDB and UniProt, and UniProt starting and ending positions matching the sequence included in the file. The latter numbering is required to include some extra residues at both the C- and N-terminus regions that might still be relevant for RNA binding.

From these RRM-RNA complexes, all protein-RNA interactions were computed using an in-house script and stored in a text file also available from the Bitbucket repository (S3 Dataset). An amino acid residue and nucleotide were considered to interact if any atom from the residue and the nucleotide were less than 5 Å away from each other. This is a broadly used interaction definition to keep strong interactions such as hydrogen bonds or electrostatic interactions, while still accounting for hydrophobic interactions that can occur at distances of 3.8–5.0 Å [32]. The analysis resulted in 13387 identified amino acid to nucleotide interactions.

### Data cleaning and alignment procedure

For the analysis, it was necessary to reduce the bias in the original complete RRM dataset. The protein sequences were extracted from the 1259 RRM domain structures (S1 Dataset) and to eliminate nearly identical RRM sequences in this set a sequence identity threshold of 99% was applied, after which 356 sequences remained in the reduced RRM dataset (S4 Dataset). This set still showed a strong bias towards certain RRM families, especially the RRM_1 Pfam family (PF00076), with 314 entries out of the 356 belonging to this family. To overcome this bias a set of 19 representative RRM domains was defined using CD-HIT [33] with a 30% sequence identity cut-off, while validating that we were selecting entries from different Pfam families, even though larger families such as RRM_1 are still repeated. This representative RRM set (S2 Table) contains very diverse RRM sequences for which the structure is available, and served as the core for generating the master multiple sequence alignment (MSA) (Fig 2).

The sequence and structure information from the reduced RRM dataset was fed into PRO-MALS3D [34] to generate the alignment (Fig 2). Besides using structure and sequence, this tool employs secondary structure predictions to generate the MSA. For our study, we used the structures of the previously selected pool of 19 representative RRMs, and the sequences for the remaining 336 domains, totalling 356 RRM sequences. After manual checks to avoid gaps in the MSA for the 6 main secondary structures (β1-α1-β2-β3-α2-β4) due to some unusually long β-strands and/or α-helices, 347 RRMs remained in the clean alignment.

The clean alignment was further enhanced to improve the often poor alignments for the loops and terminal regions. Based on the principle that for amino acids in the loop regions the most important characteristic is how they are connected to the fixed secondary structure

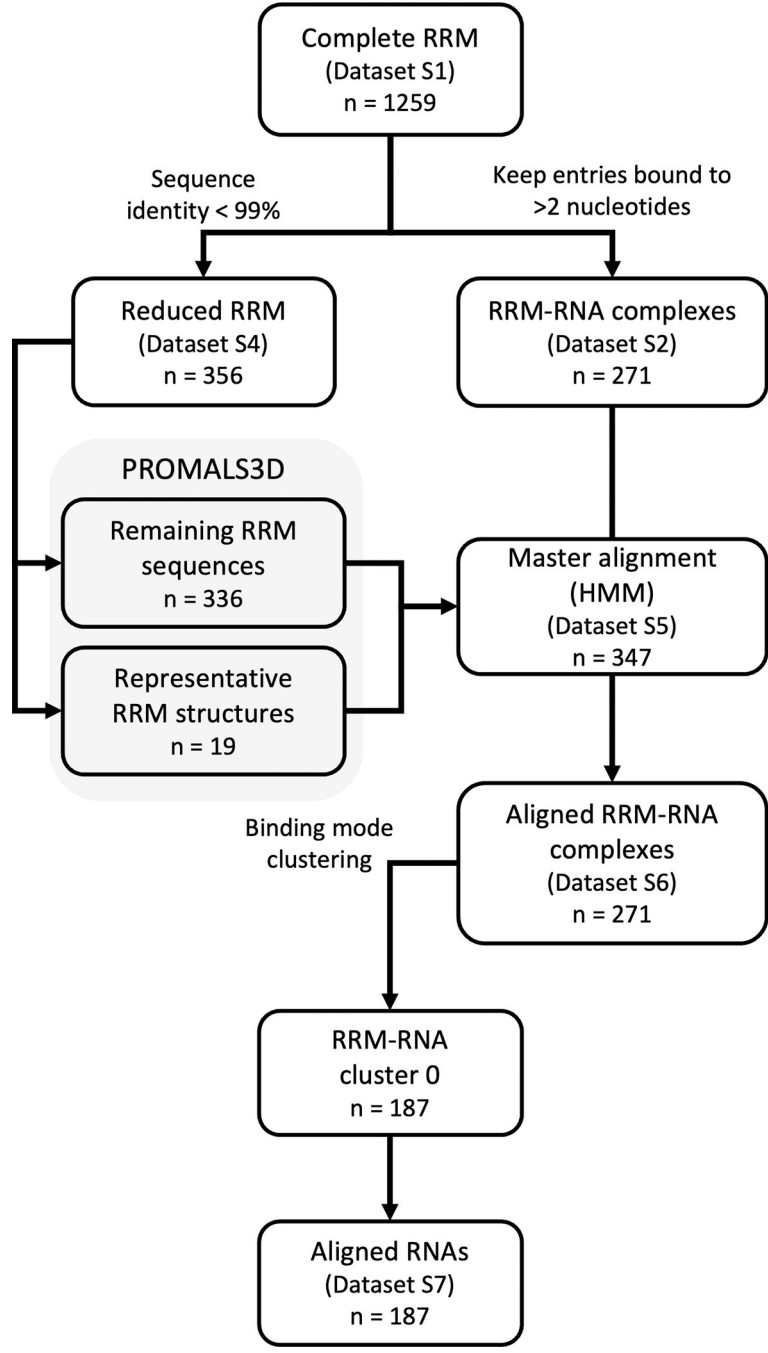

**Fig 2. Data flow diagram for the RRM structural and sequence data to generate the master alignment, use it to align the RRM-RNA complexes and cluster them depending on their binding mode so the RNAs can be aligned.** The different Datasets that have been made available are named and labelled with its corresponding number of entries.

elements, these regions were 'squeezed' so all the gaps are placed in the middle of the loop regions, or at the extremes for the N and C terminal regions, in case of shorter loops or terminal regions, respectively (Fig 3). This alignment, hereinafter referred to as the master alignment, is then used to generate an HMM useful to quickly align other RRM sequences, such as the RRM-RNA structures dataset (Fig 2). The master alignment (S5 Dataset) and the

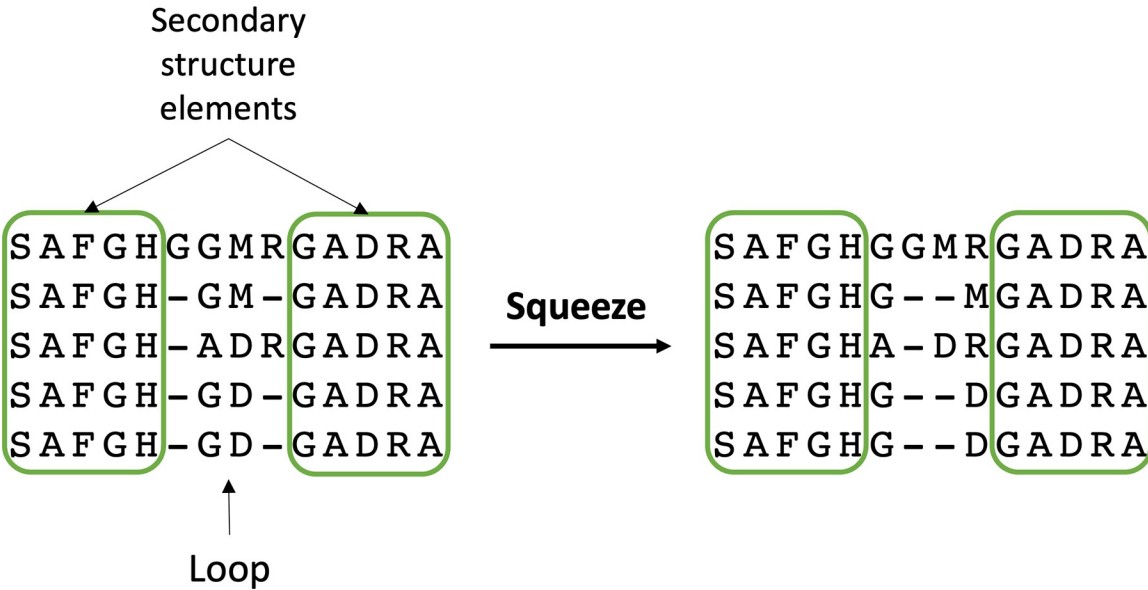

**Fig 3. Schematic representation of the sequence alignment procedure to improve the alignment in the loop and terminal regions.**

alignment for the RRM-RNA structures (S6 Dataset) are available from the supplementary information folder in the Bitbucket repository.

## RRM-RNA complexes similarity matrix

To identify and cluster the entries where RNA molecules bind in a similar orientation to the RRMs, we developed a method to compute a pairwise similarity score based on the amino acid positions of the RRM that recognize the nucleotides of the RNAs (S1 Fig). This was calculated for the 271 RRM-RNA structures available (S7 Dataset). Some entries contain long RNAs (up to 3000 nucleotides), for which the RNA sequences were truncated to keep only the part that binds to the RRM. All RNA sequences were pairwise compared, by sliding them with respect to each other and checking whether their nucleotides bind similar amino acid sequence positions in the RRM master MSA. The number of matching positions between the two RNAs (positions of the RRM that both nucleotides are interacting with) are divided by the number of unique positions between both nucleotides, i.e., the total number of different positions that those two nucleotides bind. The value of this ratio for each aligned nucleotide is added and then divided by the length of the alignment, providing a score from 0 (completely different binding mode) to 1 (the same binding mode) (Eq 1).

$$Similarity\ score = \frac{\sum_{i}^{i=n} \frac{N_{Matching\ positions}}{N_{Unique\ positions}}}{N_{Aligned\ nucleotides}}$$

For each compared RNA pair, only the alignment with the highest similarity score is retained. A similarity matrix was then constructed from these best scores that includes all the RRM-RNA complexes, and which is later used to identify the different binding modes. The matrix is available as a CSV file from the supplementary information folder in the Bitbucket repository (S7 Dataset).

To select a homogeneous cluster, we grouped all the entries that have a minimum score of 0.25 with at least 25% of the complexes in the cluster. We explored the effect of changing the

cutoffs on the cluster generation (S2 Fig), and selected a combination of values that balances the variability and the similarity within the clusters, so allowing further meaningful analysis of the complexes. The first cluster defined, hereinafter referred to as cluster 0, retained 187 entries and it was the largest found (Fig 2) using those cutoffs. Other smaller clusters were generated, but in this work, we will focus on the canonical binding mode represented by cluster 0.

After visual inspection of some of the entries within cluster 0 we verified that the RNAs were bound similarly, and that we essentially captured the canonical binding mode [1]. The entries raising the lowest similarity score were PDB Id. 6g90 (chain B–RRM1) and PDB Id. 3nnh (chain B–RRM1), with a similarity score of 0.083. Based on their best alignment from the score calculation, a UC RNA fragment from 6g90 aligns with a GU RNA fragment from 3nnh. To verify they still shared a similar binding mode both complexes were superimposed (S3 Fig) and the RMSD was calculated using the heavy atoms in the sugar for the four aligned nucleotides, obtaining a value of 1.98Å.

## RNA sequence alignment

To conduct the RNA alignment, we used the same method as for the RNA binding modes identification, based on the amino acid sequence positions of the RRM that the nucleotides of the RNA are in contact with. When comparing two RNAs, the sliding window position that generates the highest score corresponds to the best possible alignment for those sequences. To align the 187 RNAs included in cluster 0, we selected the medoid, the entry with the highest similarity scores with respect to all other entries (PDB Id. 3hhn, chain D). For the 186 remaining entries we found the best alignment against the medoid and generated the RNA MSA. Gaps were added at the 5' or 3' ends of the RNA sequences when required to ensure that nucleotides in the same position were properly aligned. This is necessary due to the different length of the RNA fragments considered, which ranges from 3 to 11 nucleotides. The FASTA file with all the RNAs aligned from cluster 0 is available from the Bitbucket repository (S8 Dataset).

## RRM-RNA scoring

The RRM-RNA scoring method we have developed, RRMScorer, is purely based on statistics derived from the carefully curated multiple sequence alignment. It is an adaptation of the GOR method [35,36] that converts the statistical information in a probabilistic framework, and which was originally used for secondary structure prediction. The GOR method applies information theory principles to calculate the information difference of the occurrence of two events. In the GOR method, the two information components calculated are 1) a specific amino acid residue type being in a particular secondary structure element and 2) that same amino acid residue type being in any another secondary structure element. The two components are calculated as a logarithm, with the background amino acid information present in both, and as the difference between them is calculated, the amino acid residue type occurrence disappears from the equation. RRMScorer relies on the same information difference equation to calculate which nucleotide-residue contacts are preferred for specific amino acid positions in an RRM. The amino-acid contribution, similar to the original GOR equation, disappears from the equation and the obtained terms result in Eq 2 below. The full development of the equation is available in Supplementary Material (S1 File). This approach is suitable for the limited amount of data and residue-level information that is currently available, as it generalises the information and avoids overinterpretation of specific RRM-RNA interactions. The method can score RNA fragments up to 5 nucleotides long, because those are the positions for which

we have sufficient information to statistically analyse.

$$I\left(\Delta N_i; R_J\right) = log\left(\frac{f_{N_i,R_j}}{f_{n-Ni,Rj}}\right) + log\left(\frac{f_{n-N_i}}{f_{N_i}}\right)$$

The scores are computed for each residue-nucleotide interacting position individually. The result is the sum of two terms; the first term computes the logarithm of the ratio between the number of times a nucleotide in position $i$ (from the RNA sequence alignment that accounts for how it binds the RRM) has been observed interacting with an amino acid residue in position $j$ ($f_{N_i,R_j}$) (based on the master alignment), over the number of times that the nucleotide interacts with any other amino acid residue ($f_{n-Ni,Rj}$). E.g., the number of times an adenine in position 1 is observed interacting with an arginine in position β1–1 is divided by the number of times adenines in position 1 interact with any other amino acid residue in position β1–1. This value is then corrected by the second term, which computes the ratio between the number of times another nucleotide is observed in position $i$ ($f_{n-N_i}$) versus the number of times the selected nucleotide is observed in that position ($f_{N_i}$). Following the previous example, this is the number of times any nucleotide except adenine is observed in RNA position 1 divided by the number of times an adenine is observed in that position.

RRMScorer takes the information from the interactions observed in all available RRM-RNA complexes in cluster 0 (training set). However, selecting all available interactions for the scoring would bias this approach to the amino acids and bound RNAs for the most studied RRMs, which are overrepresented in our dataset. To generalize the approach as much as possible, we first selected a subset of nucleotide and residue positions that interact with each other in at least 20% of the RRM-RNA complexes having different UniProt identifiers in cluster 0, so limiting the analysis to key binding positions. These retrieve all the RRM positions already well known for its importance with respect to RNA binding, such as solvent-exposed RNP1 and RNP2 residues. In total, 30 unique interactions from 20 RRM sequence positions to 5 RNA sequence positions were considered. A data frame was then generated for each of the interactions that incorporates the scores between all the residues and nucleotides in that position, as calculated from Eq 2. To get the final binding score between a target RRM sequence and a target RNA sequence, we take the average from the 30 calculated values. The matrices were rendered and coloured for all the selected interactions (E.g., S4 Fig), and are available from the Bitbucket repository under the supplementary information folder. For simplicity, we only included in the matrix amino acid residues that interact with the RNA. In summary, higher final binding scores indicate a higher overall probability that key amino acids from the target RRM have been observed to be in contact with the nucleotides from the target RNA.

### Score internal validation

To cross-validate the scores in as unbiased a manner as possible we followed several steps. First, we removed the complex for which we are calculating the score from the training set. With the remaining entries we calculated the residue-nucleotide preferences for the 30 selected interactions, then calculated the final binding score for the entry being scored, given the amino acid sequence of the RRM, and nucleotide sequence of the RNA. This constitutes the single 'actual binding' score for that entry. All the 187 entries used to generate the RRMScorer were taken as the training set, as we have experimental proof that the RNAs bind to their respective RRMs.

A randomized test set was generated to validate that the RRMScorer can identify true binders. For each of the training entries an entry was randomly generated, whereby the RRM

amino acid sequence is retained, but instead we picked a random nucleotide from the same position in the RNA alignment, but belonging to a complex with a different protein (UniProt Id). With this approach we expect to select nucleotides less likely to bind the tested protein, even though is still possible to pick the same nucleotide as in the original sequence.

## Validation with RNAcompete data

RNAcompete is an *in vitro* technique to quickly analyse the RNA binding preference of RNA binding proteins (RBP) [19]. The experimental RNA preferences for all the available RRMs were downloaded from the CISBP-RNA Database [37], resulting in information for 171 different proteins. The RNA preferences are presented as a matrix of frequencies observed for each nucleotide along 7 different positions (8 positions in a few particular cases). We reconstructed all the possible sequence combinations for those 7-mers and computed the associated bits the same way it is done for classic sequence logos [38], capturing the information content of each nucleotide in a particular position.

As RRMScorer can score up to 5 nucleotide long RNA fragments, we kept the 5-mer with the highest average bits value from the RNAcompete 7-mers. Considering that in the RNAcompete assay they used an RNA pool comprising ~240,000 short fragments (30–41 nucleotides) that guarantees that any 9-mer is at least repeated 16 times, we can assume that any 5-mer sequence is also present multiple times in the pool. Therefore, there are 1024 ($4^5$) theoretical different 5-mers, from which we can extract the associated bits value for each of the 171 RRM-containing proteins.

The 171 proteins in the dataset are divided into 3 categories based on their RBP composition. This must be done to properly align the sequences to the master alignment and analyse the resulting scores for each individual domain. There are 47 proteins that consist of a single RRM, 118 have multiple RRMs and the 6 remaining ones have at least one RRM but in conjunction with other RBPs such as KH domains or zinc fingers. The single RRMs are simply aligned to the master alignment (HMM) and the scores are calculated for all the possible RNA 5-mers (1024 sequences). For proteins with more than one RRM, we cut the sequence of each individual RRM and computed the score for the RNA 5-mers, we keep the highest score to correlate with the bits data. All the available data for each RRM domain, bits average value and RRMScorer predictions are available from the supplementary information folder in the Bitbucket repository (S10 Dataset).

## Score confidence

The confidence of the scores is calculated based on the final binding scores between the RRM and the RNA in the complex we are scoring, and how that compares with the scores from the training and randomized sets. The scores for both sets were fit into a gaussian kernel density estimator (KDE) using Scikit-learn [39]. We then can compute the likelihood for the final binding score of an RRM-RNA complex to fall on either the training or the randomized regions. The ratio between those probabilities is calculated, normalized from 0 to 1 and provided as confidence score. Low values means that the RRM-RNA complex does not show very favourable contacts according to our dataset, which places it in the predominant randomized region. Correspondingly, high values (near 1) mean that the contacts (or absence thereof) were observed very often in our training set, and we can be confident about the prediction.

## Results

### RRM alignment analysis and representation

The RRM master alignment (S5 Dataset) was validated by checking the number of gaps in the main secondary structure elements, and comparing the sequence logos with prior information

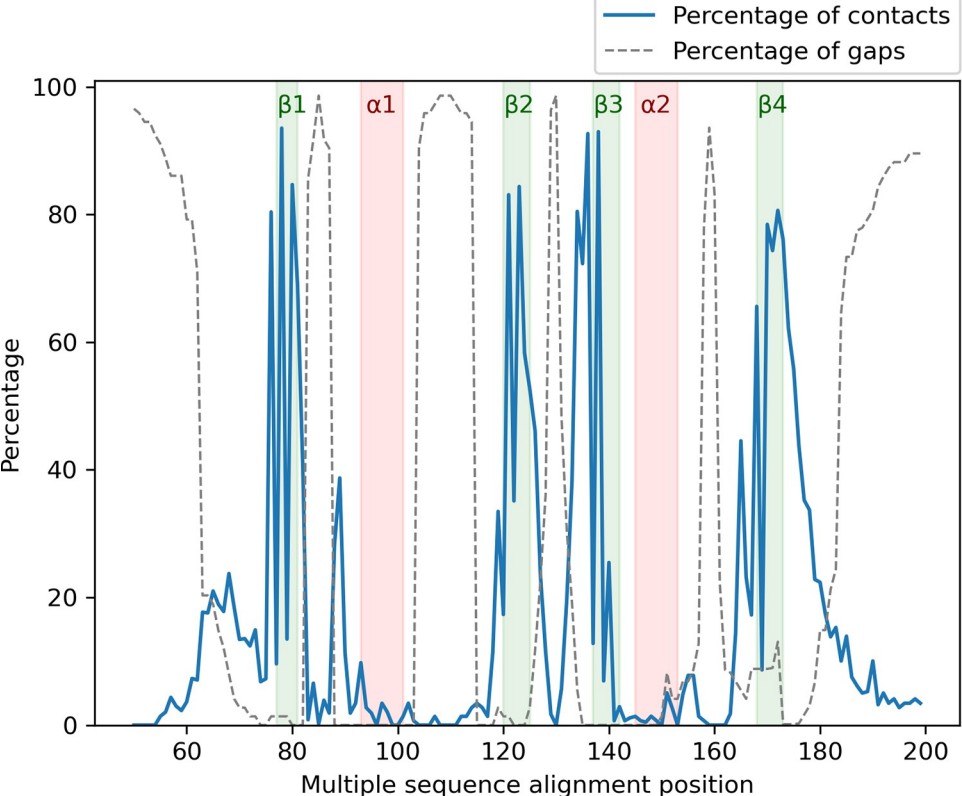

**Fig 4. Percentage of contacts (Blue line) and gaps (grey dashed line) for all the positions of the 271 bound RRM alignment.** The β-strands and α-helices are depicted and labelled in green and red, respectively.

about the β3-strand and β1-strand, also known as ribonucleoprotein domain 1 (RNP1) and RNP2 [1], respectively. The alignment for the RRM domains from the RRM-RNA complexes (S2 Dataset) was similarly validated, and the interactions previously extracted for all the complexes (S3 Dataset) were mapped into the alignment to analyse the RNA contacts along the protein structure. The frequency of the contacts and gaps is calculated as a percentage for each of the positions in the alignment of the 271 RRM-RNA complexes (Fig 4). As expected, most of the contacts occur in the β-strands, the 1st, 3rd and 5th loop, and the C-terminal region, which further validates the RRM-RNA alignment. The gaps are concentrated in the middle of the loops because of the alignment curation (see M&M).

To better identify the different RRM sequence positions from the RRM master alignment, we created a cartoon representation of the RRM showing the most relevant sequence positions in relation to the conserved structural features (Fig 1B). Only the positions of the canonical RNA binding interface are depicted by individual spheres for simplicity. The position's labelling is designed in a grid-based system, e.g., position β1–1 in Fig 1B. With such a representation it is easier to refer to the different RRM positions, which becomes particularly useful when comparing different RRMs or when analysing to which positions the RNAs bind. The light-green and dark-green colours for the β-sheet positions indicate exposed or buried residues, respectively. Positions with a significant number of observed RNA interactions (contacts observed in more than 20% of the proteins with different UniProt identifiers) are highlighted in red. As expected, the exposed positions of the β-strands are the ones most often interacting with RNA. These positions include the well-known conserved aromatic residues in β1–3, β3–3 and β3–5 that usually anchor the RNA by pi stacking (positions labelled in Fig 1B) [1].

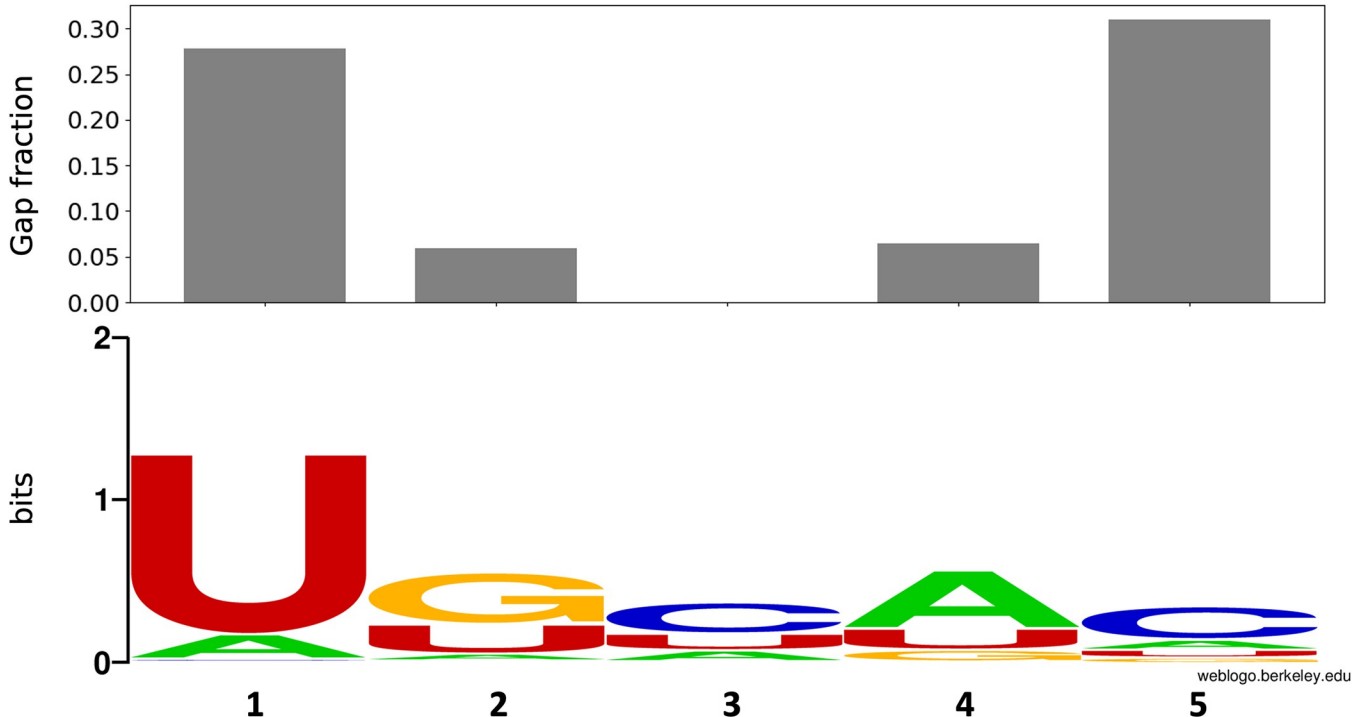

**Fig 5.** Gap fraction for the 5 core RNA positions in the alignment used for the scoring (top) and nucleotide conservation (bottom). Sequence logo generated with WebLogo [38].

## Selecting the RNA binding mode and aligning the RNAs

To connect the RRM alignment with RNA recognition, the RNA sequences were aligned to compute how often a residue in a specific RRM position interacts with a nucleotide in a specific sequence position of the RNA. Considering that the RNA can adopt different conformations upon binding the RRM, we clustered the structures of the RRM-RNA complexes in subgroups with comparable positions of the bound RNAs with respect to the protein.

For the 187 complexes grouped in cluster 0, the RNAs bind to the RRMs in a very similar way, thus allowing to align the RNA sequences in relation to the RRM. In general, the number of interacting nucleotides in the complexes is variable, but most of the RRMs interact with 2 to 5 different nucleotides [3], even though there are some RRM domains with non-canonical binding that can interact with up to 8 nucleotides [40,41]. Based on this information, we analysed the fraction of gaps for the different positions in the RNA alignment. Gaps must be included to properly align RNAs based on the positions of the RRM they interact with (E.g., aligning UAGU with GUAGU RNA motifs, and assuming they bind their respective RRMs in the same way, a gap must be added to the 5' end of the shorter RNA motif so it aligns with the latter). After analysing the aligned complexes, we observed that the 5 positions with fewer gaps (Fig 5 top) are recognized by the central two β-strands (β1 and β3, Fig 1) of the RRM. We defined this region as the core of the RNA alignment, and focused on this for further analysis. The nucleotide conservation for these positions was also determined (Fig 5 bottom). Notably, although we are restricted to the canonical binding mode, a wide range of RNA sequences is covered. The only position where less sequence variability was captured is the 1st position of the RNA alignment. This is mostly due to a higher percentage of gaps and to a bias towards two RRM proteins, SNRPA (UniProt Id. P09012) and U2AF (UniProt Id. P26368) with 69 and 23 entries in cluster 0 respectively, that often bind a uracil in this position.

### RRM-RNA scoring

The RRM-RNA scoring method we developed, RRMScorer, predicts how likely it is for a given RNA sequence to bind a target RRM sequence. It is specifically developed for the canonical binding mode (captured by the RRM-RNA complexes in cluster 0) and it uses both the RRM and RNA alignments to generate a score to estimate RNA binding. The score is individually calculated for the 30 selected RRM-RNA contacts and then averaged. Positive contact scores indicate that a specific amino acid-nucleotide interaction is likely to be encountered, while negative scores have the opposite meaning. Scores close to 0 mean that there is no clear preference, e.g., the conserved aromatic residues in RNP2 (RRM position β1–3) and RNP1 (RRM positions β3–3 and β3–5) are barely specific for any nucleotide (S4 Fig), which is coherent with the fact that pi stacking interactions are not nucleotide-specific.

Note that the current analysis of RRMScorer is restricted to RRMs in cluster 0 and assumes that the RNA binding mode does not change. Significant protein sequence variations might change the RNA binding mode and will require further analysis. Moreover, due to the limited size of the training set, not all possible amino acid-nucleotide contacts are sampled, and thus scoring of RRM-RNA complexes with interactions that have not been observed before is less reliable. The unbiased number of observed contacts in the training set that is used to calculate the scores is also shown in the preference matrices (E.g., S4 Fig), below each of the scores. This value is the sum of the contact conservation but UniProt entries normalised, e.g., if a specific contact is observed in 8 out of the 10 available structures for the same UniProt entry, this contributes with 0.8 to the unbiased number of observed contacts. Following this procedure, we avoid the bias towards overrepresented protein structures in the dataset. When this value is absent, no such residue-nucleotide contact was observed in our training set for those positions. Residues that do not contact any nucleotide in a specific position are not displayed for simplicity.

### Scoring validation

To validate RRMScorer, its performance was assessed on different independent experimental datasets to certify that the method is capable of predicting the binding capabilities of the RRMs.

### Internal validation

We performed an internal validation by computing the scores for the entries in cluster 0, referred to as the training set, and comparing them with the scores from a set of randomly generated RNA sequences, referred to as the randomized set. The scores were calculated as explained in the score validation section on Materials and methods, taking out the entry from the dataset before computing the matrices with the scores. Following this procedure, we ensure that we are not biasing the scoring.

The distribution of the scores for the training and randomized entries (187 entries in each set) are compared and plotted (Fig 6). A separation between the sets is observed confirming that the scores can discriminate with certain confidence between the RNA sequences that would bind to an RRM from the ones that would likely not bind. The internal validation results with the training and randomized scores are available from the Bitbucket repository under the supplementary information folder (S9 Dataset).

### Validation with RNAcompete data

Using the RNAcompete data we were able to evaluate the RRMScorer method with a large set of experimental RNA binding preferences. The data were processed to translate the RNA

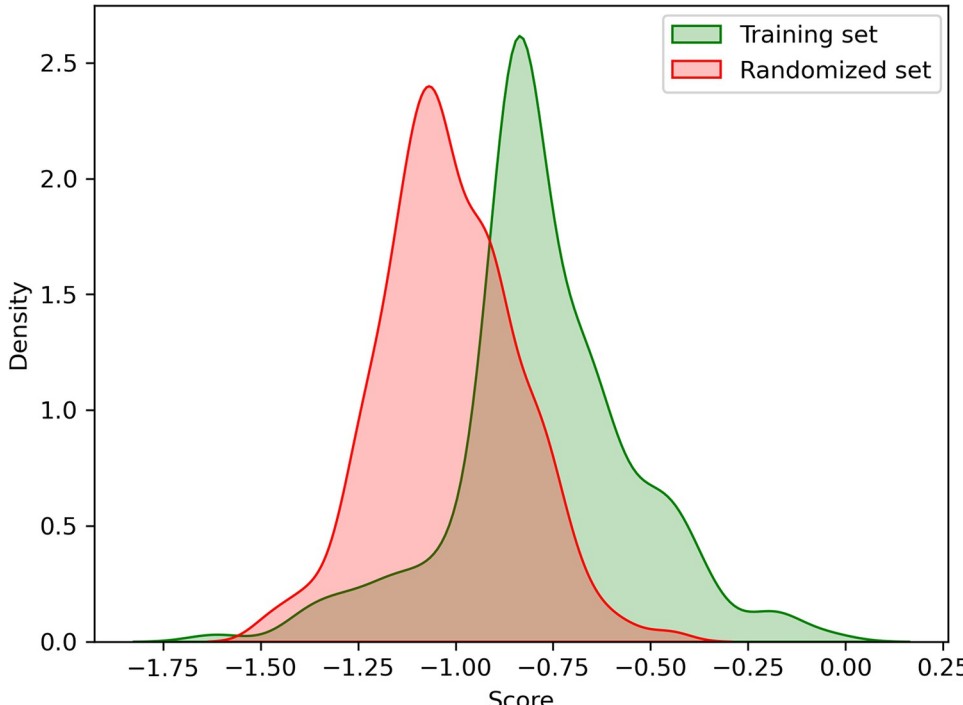

**Fig 6. Score distribution for the training set (scores from experimentally solved RRM-RNA complexes) and randomized set (randomly generated RNAs).**

binding preferences into bits values that then we can compare with our predictions (S10 Dataset). RNA fragments with a higher average bits value should be better binders for that particular RRM, and consequently should correlate with higher scores from our method.

The correlations between the scores and bits values are shown for the three different categories (Fig 7). While we can make a one-to-one connection between the nucleotide preferences and the single RRMs, in case of multiple RRMs, and especially when other RBPs are present, the observed preferences might be a combination of different specificities of each individual domain. In fact, in general the highest affinity domain interaction will dominate results, and domain-specific contributions are in general not known. This fact is also represented on the different correlation of the medians observed for single RRM, multiple RRM and multiple RBP categories, with respective Pearson correlation coefficients of 0.78, 0.30 and 0.07. The clear correlation between the single RRMs bits values and RRMScorer predictions further validates our method, making it particularly useful for genome scale studies and large-scale screenings of RNA candidates.

### Validation with Musashi-1

Musashi-1 (MSI1) is a protein containing two tandem RRM domains that is involved in post-transcriptional regulation processes, controlling target mRNA transit and translation [42]. It is expressed in several species, and in humans its malfunction is often associated with cancer development [43]. As a relevant RRM use case, we compared the experimental information available about the RNA binding affinity with the predictions from RRMScorer. Zearfoss *et al.* analysed several RNA mutants to define the Musashi RNA binding specificity [12]. They mutated all the positions of a 12-nucleotide long RNA containing a motif previously identified by SELEX [13], and determined that MSI1 RRM1 (UniProt Id. Q61474) specifically recognizes

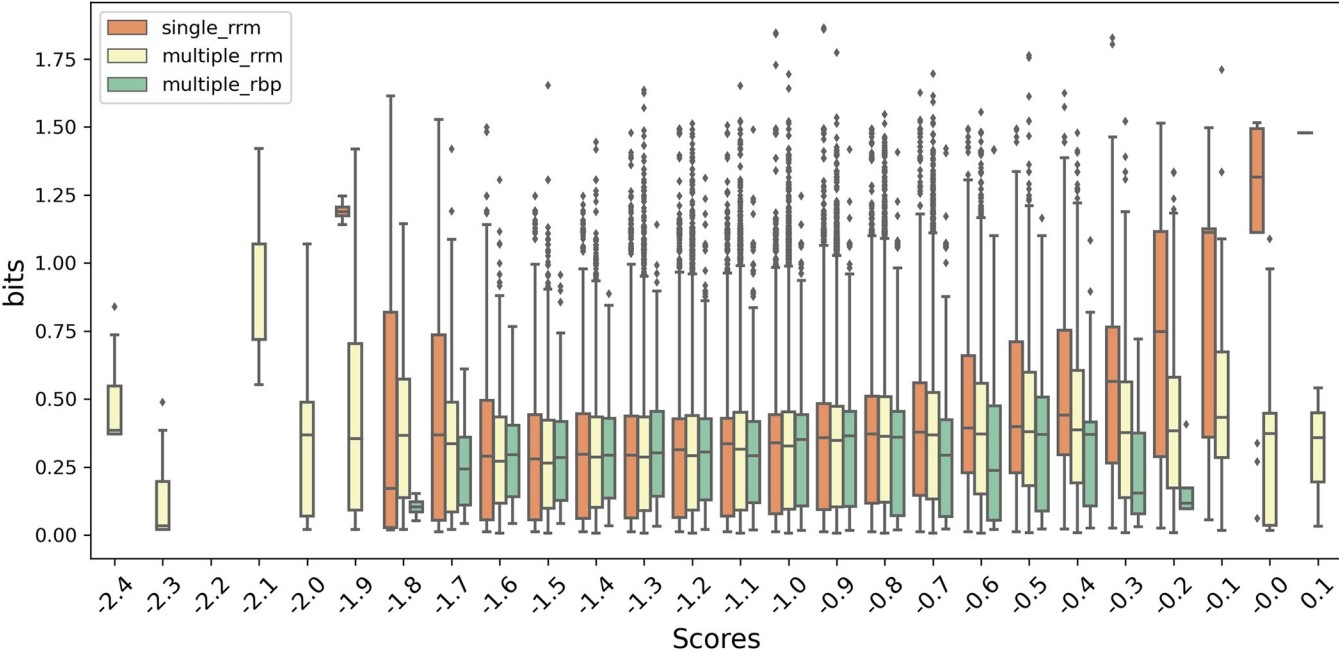

**Fig 7. Correlation between the bits values derived from the RNAcompete data and the scores obtained with RRMScorer.** For a clearer depiction of the single RRM (orange), multiple RRM (yellow) and multiple RBP (green) categories, each box corresponds to the distribution of the bits values for a range of 0.1 in the score axis.

a core motif of 3 nucleotides, UAG. If this core region is mutated, the binding affinity drops, while mutating other positions of the RNA does not have a substantial impact on the binding.

The authors determined the binding affinities between the RNA mutants and the MSI1 RRM1 using fluorescence polarization assays. The mouse variant was used on the experiments, which is identical to the RRM1 human variant. We then used RRMScorer but removed the MSI1 RRM1 entry from the training set to make the test-case as agnostic as possible. For this validation we used a window size of 3 nucleotides to calculate the scores, as the authors claim that this is the length that MSI1 specifically recognizes. We correlated the experimental results with the 3-mer fragment raising the highest score for each of the 36 RNA mutants (Fig 8). A table with the scores and affinity data is available from the supplementary information file (S3 Table).

The correlation proves that RRMScorer successfully distinguishes between high affinity RNAs containing the UAG core motif (bottom right corner) that are in the low nM range (40–200 nM), from the ones without the conserved three-nucleotide motif, whose affinities drop to the μM range (1000–2500 nM). The guanine in the UAG motif is the most relevant nucleotide with respect to the binding affinity, when mutated to any other nucleotide the resulting interaction is on average 37-fold weaker than with the WT RNA (S3 Table). Using as a reference the PDB entry 2rs2 for MSI1 RRM1, two residues are involved in the specific recognition of this guanine, K21 and F65, interacting with their respective sidechains (Fig 9A). The corresponding positions in the RRM alignment for K21 and F65 are β1–1 and β3–3, respectively (Fig 1B and S6 Dataset), and the guanine from the UAG motif corresponds to the RNA position 4 in the RNA alignment (Fig 5 and S8 Dataset). In agreement with the experimental observations, a lysine in position β1–1 only shows a positive score for guanine (score of 0.48), and negative scores when binding any other nucleotide (-0.84, -1.64 and -0.58 for adenine, cytosine and uracil, respectively, Fig 9B). On the other hand, a phenylalanine in position β3–3 does not

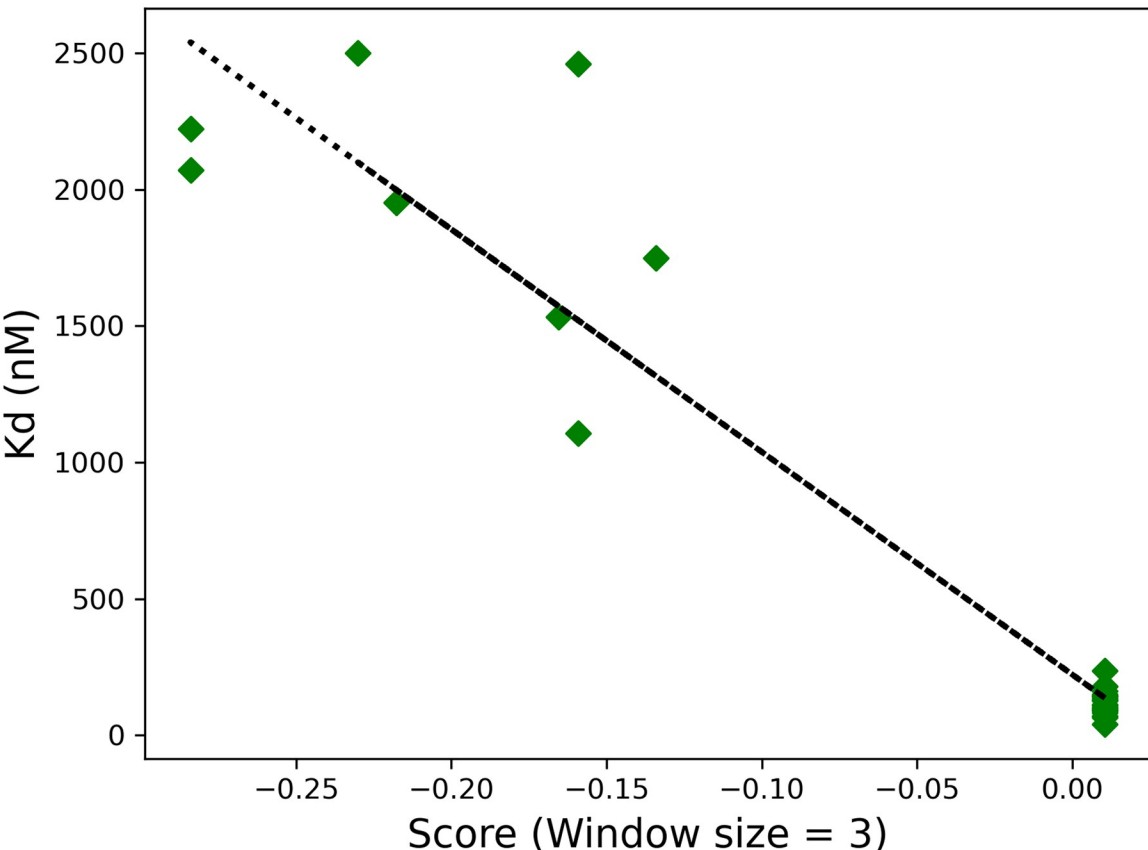

**Fig 8. Experimental Kd values-scores correlation for the 36 RNA mutants tested against MSI1 RRM1, using a window size of three nucleotides as it is the length that this RRM specifically recognizes [12].**

show a preference for any nucleotide (Fig 9C). This was expected as pi stacking interactions are not nucleotide-specific.

By computing the scores for all sliding windows in the RNA to find the highest possible score, we also predict which part of the RNA is more likely to bind the RRM. This is particularly useful to spot likely binding regions in long RNAs and the prediction confidence (Fig 10). The 5-nucleotide window with the highest score is clearly separated from the other RNA fragments and predicted with high confidence, and it agrees with the SELEX consensus defined for this protein (G/A)U1–3AGU [12,13].

### Validation with SRSF1

The complex of the human prototypical SR protein SRSF1 RRM1 with RNA was not available when we generated the master alignment. Therefore, it was not used to develop RRMScorer and it can be used as an external validation test case. The structure was recently released as part of a work by A. Cléry *et al*. [24] where the authors identified that this RRM has a strong preference to bind cytosines (PDB Id. 6hpj). They also engineered a variant to gain the ability to bind uridines, mutating a single residue in the β4 strand from glutamate to asparagine (E87N).

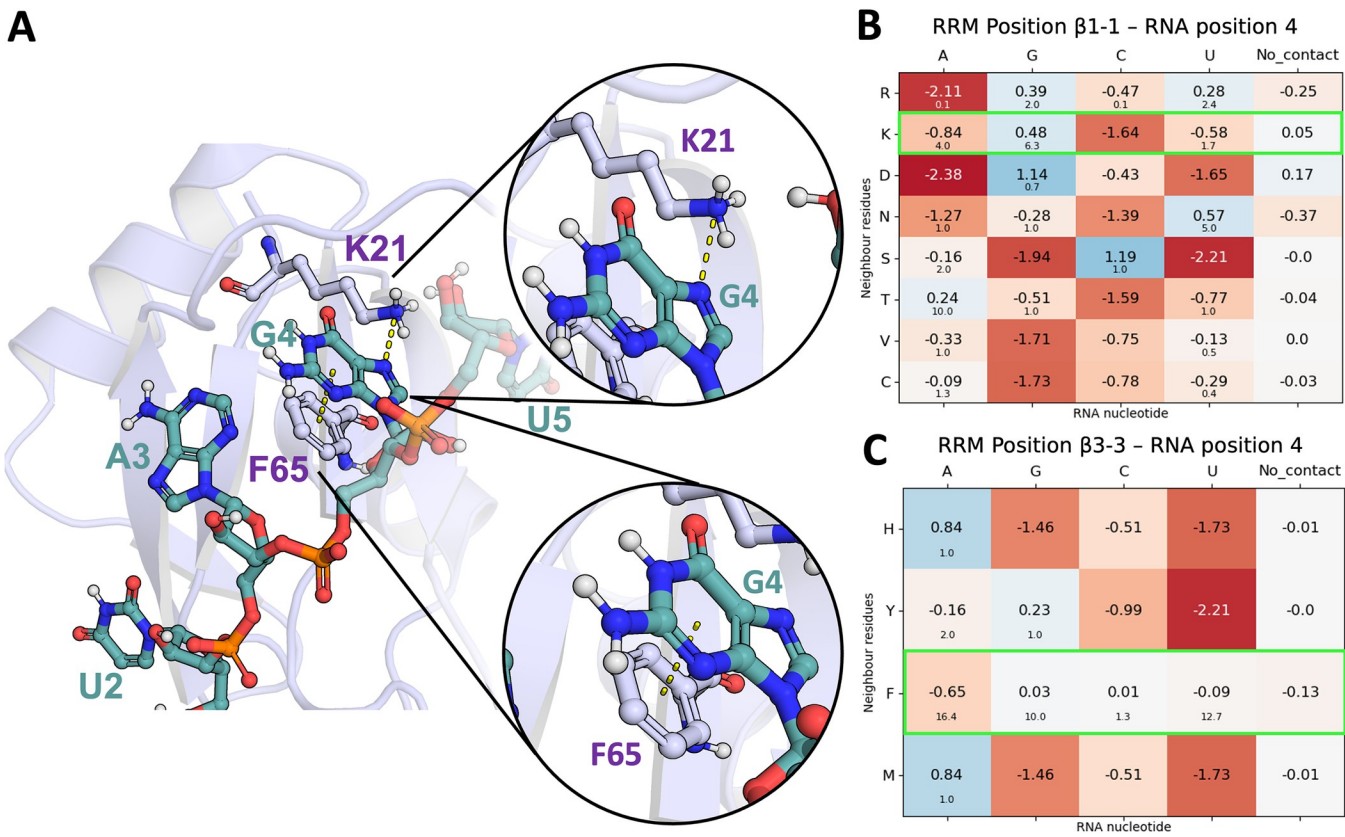

**Fig 9.** A) Cartoon representation of the MSI1 RRM1 protein in complex with GUAGU (G1 not shown, PDB Id. 2rs2). The protein backbone is shown in light blue and heavy atoms are shown in red (O atoms), blue (N atoms), Orange (P atoms), light blue (C atoms of RRM) and sea blue (C atoms of RNA). The RNA nucleotides and the residues involved in the specific recognition of G4 (K21 and F65) are shown as sticks and spheres. B) and C) Score matrices for the alignment positions matching the interactions between K21 and F65 with G4, respectively (highlighted in bright green).

We wanted to assess whether our method can separate the RNAs that bind from the ones that do not bind, and whether it can capture the difference in binding of the engineered variant. The closest RRM sequence in cluster 0 has 51% sequence identity (SRSF3 RRM1), but a visual inspection of the complex shows a similar binding mode, which is crucial for the reliability of RRMScorer. The SRSF1 sequence is then aligned with the rest of the entries in the cluster so we can compute the scores for this entry and the mutated variant. For this step, only the HMM was used, not any structural information, which our approach does not require.

The authors tested the WT SRSF1 RRM1 binding with poly-A, poly-G, poly-U and poly-C by NMR, with only the latter binding the protein as validated through chemical shift perturbations in 1H-15N HSQC spectra. Our scores showed the same trend, while poly-A, poly-G and poly-U yield scores of -0.95, -1.00 and -0.91 respectively (Table 1), which locates them in the predominant randomized region (Fig 6), poly-C gets a score of -0.59, which clearly falls in the training set region or likely-binder region. Notably, the engineered variant E87N increases the poly-U score from -0.91 to -0.82 (Table 1), which is a significant change considering that only one residue is mutated from the 20 RRM positions used for the scoring. This score now falls in the likely-binder region (Fig 6) and agrees again with the NMR assays where the authors determined that the RRM1 mutant can bind uridines. The corresponding alignment position for E87 is β4–3 (Fig 1A) and it interacts with a cytosine in position 4 based on our RNA alignment for cluster 0 (Fig 5). A close look at this interaction (Fig 11A) shows how the carboxyl group of

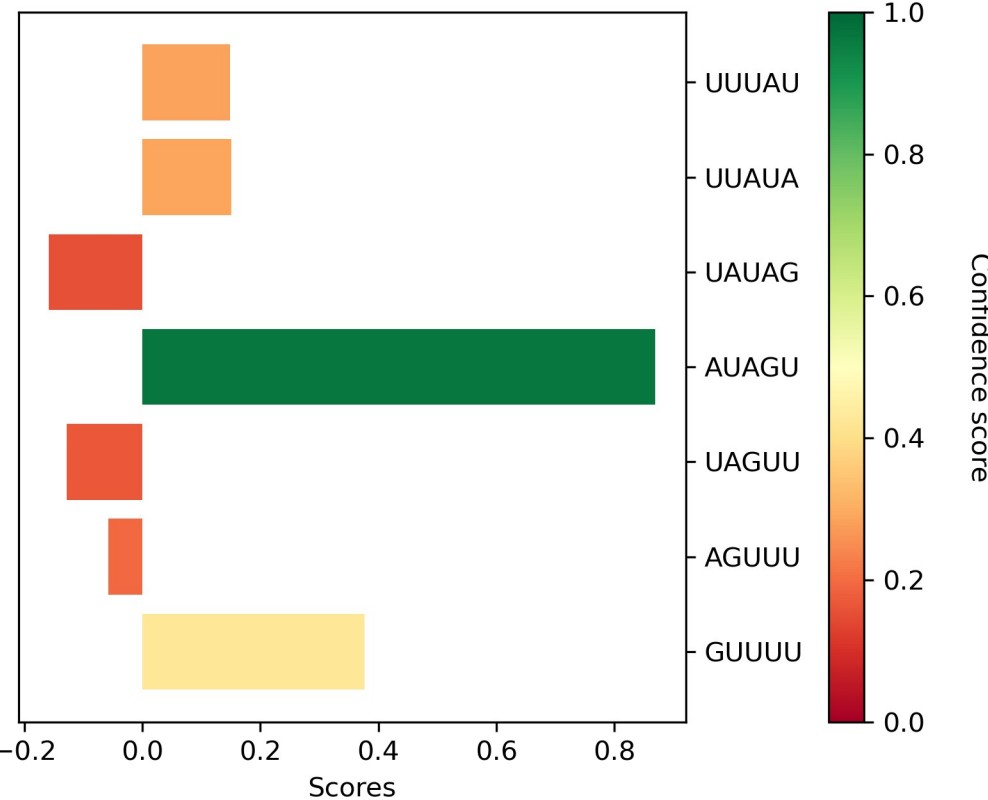

**Fig 10. Scores for the 5-nucleotide sliding windows of the WT RNA tested by N. Ruth Zearfoss *et al*.** The bars are coloured depending on the score confidence, that goes from 0 to 1 for no-confidence to maximum confidence predictions, respectively. To ease the interpretation of the results, we add 0.89 to each of the scores obtained as it is the value that better separates the training and randomized regions according to the receiver operating characteristic curve (ROC curve). Therefore, positive scores correspond to likely binders while negative scores correspond to RNA fragments less likely to bind the RRM.

the glutamate sidechain interacts with the amino group of the cytosine, while not being able to recognize a uracil in this position, as stated by the authors [24]. The single point mutation E87N that enables this RRM to bind a uracil is in perfect agreement with our scores (Fig 11B), in fact, an asparagine is the only residue showing a positive score for uracil (from the residues observed in our training dataset).

## Discussion

We developed RRMScorer, a novel scoring method to estimate RRM-RNA binding from sequence information only. RRMScorer provides scores for the probability that a given RNA

**Table 1. Scores for the 4 tested RNAs by A. Cléry *et al*. coloured in green for the RNAs that bind the target on their NMR assays and in red for the RNAs that do not bind. The symbols reflect the score change after the E87N mutation.**

|       | SRSF1 (WT) | SRSF1 (E87N) |
|-------|-----------|--------------|
| PolyA | −0.95     | −1.02 ↓      |
| PolyG | −1.00     | −0.98 ≈      |
| PolyU | −0.91     | −0.82 ↑      |
| PolyC | −0.59     | −0.69 ↓      |

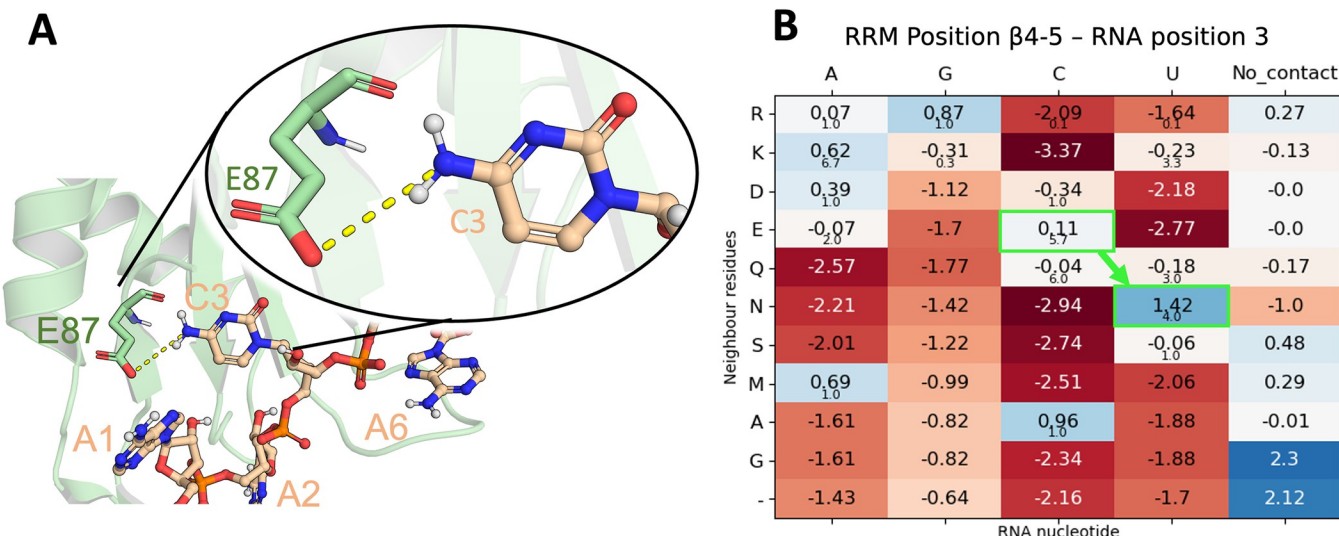

**Fig 11.** A) Cartoon representation of the SRSF1 RRM1 protein in complex with AACAAA (PDB Id. 6hpj). The protein backbone is shown in pale green and heavy atoms are shown in red (O atoms), blue (N atoms), Orange (P atoms), pale green (C atoms of RRM) and pale orange (C atoms of RNA). The RNA nucleotides and the residue involved in the specific recognition of C3 (E87) are shown as sticks and spheres. The interaction between E87 and C3 is shown as yellow dashed line. B) Score matrix for the respective alignment position of the interaction between E87 and C3. The change in score for the E87N mutation performed by Cléry *et al*. is highlighted in bright green.

sequence binds to an RRM protein and was validated on both computational and experimental data. RRMScorer does not predict precise binding affinities, but rather provides relative scores to compare different RNA sequences in relation to a particular RRM, or to compare the effect of different amino acid mutations in a particular RRM. We also focus solely on the canonical binding mode, therefore, scores predicted for RRM proteins not belonging to the analysed cluster are less reliable. Notably, our method has proved successful with SRSF1 RRM1, which falls within the canonical binding mode [24] but was not included in our original analysis. RRMScorer is also consistent with the RNAcompete large scale experimental data, especially for the single RRMs when there is a one-to-one connection between the RRM and the obtained RNA preferences, which does not apply for multiple RRMs where several RRMs contribute to the RNA preferences. The performance of our method makes it very useful on genomic scale studies to find good RNA candidates for a specific RRM. This could also then be coupled with state-of-the-art methods to predict the structure of the RNA-RRM complex, such as RoseTTA-FoldNA [30].

The generation of the RRM master alignment is one of the key and non-trivial steps of this work. While the RRM fold is highly conserved, its sequence has experienced extensive changes across evolution to modify RRM sensitivity and specificity towards different RNA molecules. This has led to the broad range of functions that this protein motif performs. Considering the low sequence identity within the set, alignment methods purely based on sequence were not successful. Purely structure-based methods did not produce the expected results either, for example due to the different length of some of the beta-strands many gaps were included within some of them. The combination of sequence and structure is essential, with in our case PRO-MALS3D [34] generating the best alignment from the tools we tried. In addition, the rearrangement of the loop residues in the alignment with respect to the secondary structure elements was essential to better capture the RRM amino acid preferences in relation to RNA binding.

Our alignment captures as much sequence variability as possible, while avoiding bias towards overrepresented families and keeping a high-quality alignment. This allows us to

extract reliable preferences between the protein residues and the nucleotides. We decided to build our scoring method on statistics derived from the master alignment and available structures to get a better grip on the data. Other methods based on machine learning would have provided relevant insights as well, but their interpretation would be less straight forward, and likely less generic. The limited number of available RRM-RNA complexes is also an impediment for most machine learning algorithms. With our statistical approach we can score any RRM-RNA complex while easily tracking the individual scores of each residue-nucleotide pair. This information can be useful to rationally design new RRMs.

Still, limited data availability is the main reason why our method is restricted to the canonical binding mode. We generated clusters for other binding modes but the number of complexes available was very limited, which makes it difficult to define well-grounded binding preferences. Finding validation sets was also challenging, because big changes on either the protein or the RNA sequence can completely change the RNA binding mode, so invalidating use of our method. From the protein structure side, the data availability is no longer a limitation after the AlphaFold Protein Structure Database release [44]. Even though it does not solve the RNA recognition problem [45], current challenges purely based on protein structure, such as assessing the preferred RNA binding mode of an RRM, might be solved soon, although the current inability of such methods to cover dynamics and multiple conformations remains a bottleneck to be solved. The more generic approach we present here might therefore be more applicable at this point in time; by 'averaging' the limited available information we reduce complexity but enable interpretability. The amino acid representations extracted by unsupervised language models [46] might in this context also be able to provide generalisations of amino acid behaviour that are applicable for improved prediction of RRM-RNA binding.

## Supporting information

**S1 Fig. Schematic procedure for the similarity score calculation between two RNAs based on their binding with the RRM.** The two RNAs are aligned in all possible combinations using a sliding window, with the number of matches and unique positions each nucleotide interacts with in the RRM sequence counted for the 4 positions aligned in this example. The ratios between the matches and unique positions are added and then averaged for all the positions by dividing them by the length of the alignment.
(PDF)

**S2 Fig.** Variation on the number of clusters (A) and number of entries in the biggest cluster (B) depending on the chosen cutoffs for similarity score (X-axis, scores from 0 to 1) and percentage of entries (Y-axis from 0% to 100%) that should have an equal or higher similarity score with the rest of the cluster. The chosen cutoff for the cluster generation is highlighted in yellow.
(PDF)

**S3 Fig. Superimposed RRM-RNA complexes with the lowest similarity score in cluster 0, PDB Id. 6g90 (chain B, green) and PDB Id. 3nnh (chain B, orange).** The aligned nucleotides used for the RMSD calculations are labelled.
(PDF)

**S4 Fig. Scores for the conserved aromatic positions in RNP2 (β1–3) and RNP1 (β3–3, β3–5) in contact with their respective RNA positions (phenylalanine and tyrosine scores are highlighted in bright green).**
(PDF)

**S1 Table. PFAM identifiers and related metadata of the selected RRM families for the analysis.**
(PDF)

**S2 Table. Identifiers of the 19 selected structures to use in PROMALS3D.**
(PDF)

**S3 Table. Experimental Kd values correlated with the RRMScorer scores for the WT RNA and the 36 RNA mutants tested against MSI1 RRM1, using a window size of three nucleotides.**
(PDF)

**S1 File. S1–S4 Equations: Basis for the development of the RRMScorer equation.** Our method is based on the information difference between the occurrence of two events, in our case the information of how often a specific nucleotide interacts with a specific residue, $I(N_i; R_J)$, and the information when that same nucleotide interacts with any other residue, $I(n - N_i; R_J)$ (S1 Equation). The individual terms are developed in S2 and S3 Equations where: $f_{N_i,R_j}$ is the number of occurrences for a specific contact between nucleotide $i$ and residue $j$; $f_{R_j}$ the number of times residue $j$ is in that position; $f_{N_i}$ the number of times nucleotide $i$ is in that position; $f_{n-N_i,R_j}$ the number of times nucleotide $i$ interacts with any residue but residue $j$; $f_{n-N_i}$ the number nucleotides other than nucleotide $i$ in that position; R the total number of residues in the dataset. As it's a difference between the two logarithms, the common terms that account for the number of specific residues in position $j$ ($f_{R_j}$) and total number of residues in the dataset (R) disappear from the equation. After the simplification we obtain S4 Equation (Same as Eq 2 in the main text).
(PDF)

**S1 Dataset. Complete list of the RRM structures retrieved from the PDB, with the RRM sequence and the sequence range for both PDB and UniProt.**
(FASTA)

**S2 Dataset. List including the identifiers for the RRM-RNA complexes interacting with three or more nucleotides.** The identifier name is organized as follows: <UniProt ID>_<RRM number>_<PDB ID>_<protein chain>_<PDB numbering>_<UniProt numbering>_<Internal numbering for mapping>_<RNA chain>.
(TXT)

**S3 Dataset. Contacts list for all the RRM-RNA complexes.** Each line corresponds to a specific contact and it is organized as follows: <PDB ID>_<Protein chain>, <PDB residue number>_<Residue one letter code>_<Nucleotide one letter code>_<PDB nucleotide number>_<RNA chain>.
(TXT)

**S4 Dataset. Fasta file with the reduced RRM dataset after applying a 99% sequence identity threshold.**
(FASTA)

**S5 Dataset. RRM sequence alignment in fasta format for the 347 RRM selected sequences. Generated using PROMALS3D.**
(FASTA)

**S6 Dataset. RRM sequence alignment in fasta format for the 271 RRM-RNA complexes.** Generated via HMM from the master alignment.
(FASTA)

**S7 Dataset. RNA binding similarity matrix for the 271 RRM-RNA complexes.** Values close to 1 refer to similar biding modes while close to 0 correspond to different binding modes. A score of 0 is given when an RRM-RNA complex is compared with itself. The matrix is available in csv format.
(CSV)

**S8 Dataset. Fasta file with the RNA alignment for the 187 RRM-RNA complexes included in cluster 0.**
(FASTA)

**S9 Dataset. CSV file with the internal validation results for the 187 RRM complexes in cluster 0.**
(CSV)

**S10 Dataset. JSON file with the protein identifiers, RRM domain, bits value from RNA-compete data and RRMScorer predictions.**
(JSON)

## Acknowledgments

We thank Dr David Bickel for his help on protein structure visualization and Adrián Diaz for the IT support.

## Author Contributions

**Conceptualization:** Joel Roca-Martínez, Michael Sattler, Wim F. Vranken.

**Data curation:** Joel Roca-Martínez, Hrishikesh Dhondge.

**Formal analysis:** Joel Roca-Martínez.

**Funding acquisition:** Wim F. Vranken.

**Investigation:** Joel Roca-Martínez, Hrishikesh Dhondge.

**Methodology:** Joel Roca-Martínez.

**Project administration:** Wim F. Vranken.

**Resources:** Wim F. Vranken.

**Software:** Joel Roca-Martínez, Hrishikesh Dhondge.

**Supervision:** Wim F. Vranken.

**Validation:** Joel Roca-Martínez, Hrishikesh Dhondge, Michael Sattler.

**Visualization:** Joel Roca-Martínez.

**Writing – original draft:** Joel Roca-Martínez, Wim F. Vranken.

**Writing – review & editing:** Hrishikesh Dhondge, Michael Sattler, Wim F. Vranken.

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
