## [Decision Letter · Decision Letter 0]

24 Sep 2022

Dear Prof. Dr. Vranken,

Thank you very much for submitting your manuscript "Deciphering the RRM-RNA recognition code: A computational analysis" for consideration at PLOS Computational Biology.

As with all papers reviewed by the journal, your manuscript was reviewed by members of the editorial board and by several independent reviewers. In light of the reviews (below this email), we would like to invite the resubmission of a significantly-revised version that takes into account the reviewers' comments.

We cannot make any decision about publication until we have seen the revised manuscript and your response to the reviewers' comments. Your revised manuscript is also likely to be sent to reviewers for further evaluation.

Sincerely,

Shi-Jie Chen

Academic Editor

PLOS Computational Biology

Nir Ben-Tal

Section Editor

PLOS Computational Biology

Reviewer's Responses to Questions

**Comments to the Authors:**

Reviewer #1: Roca-Martinez and coworkers have performed a computational analysis of RNA recognition motifs (RRMs) and RRM-RNA complexes in an effort to develop a scoring method for predicting and evaluating the probability of interaction between canonical RRMs and single-stranded RNA. The authors use available sequence and structure information to obtain individual scoring matrices for commonly observed interacting positions in RRM and RNA sequences (identified through multiple-sequence alignments), describing the preference of different nucleobase types to interact with different residue types. While the question of understanding the physicochemical underpinnings of RNA-protein interactions and predicting and sculpting their sequence determinants is extremely timely and important, the comments below should be addressed in detail before the suitability of the manuscript for publication can be adequately assessed.

Major comments

1. The description of the RRM-RNA scoring approach (p. 6), the very heart of the manuscript, is unclear and sloppy. Equation 2 is not consistent with the text and a proper explanation of the symbols and indices used is missing (denominator in the first term different from text, fn not explained, index i in denominator in the first term different from index J in the text etc.). Also, the explanations are given in an incomplete way e.g. the sentence “…is related to the number of times adenines interact with any other amino acid residue in position beta1-1” is missing the crucial qualifier “adenines at position 1”. This makes it hard to comprehend how the scores were actually calculated.

2. More importantly, the motivation and the physical foundation of the scoring function is not adequately explained. Centrally, the scores do not consider the frequency of amino-acid residues observed at a specific position (see e.g. the second term in Eq. 2), making it not symmetric when considering nucleotides and residues, respectively. In the example given on p. 6, the score should depend on the frequency of arginines interacting with adenines as well as with other nucleotides, but this is not included.

3. Also, the scoring function shares resemblance with the standard quasi-chemical approach for defining knowledge-based potentials (Miyazawa, S. and Jernigan, R.L., Macromolecules, 18, 534-552 (1985)), but with important differences. Namely, the authors here normalize the number of occurrences of a given event (e.g. presence of a nucleotide at a given site interacting with a given residue) by the number of all events other than that event (e.g the number of interactions of that nucleotide with all the other residues, except the one in question) and not the total number of all events (e.g. the number of interactions of that nucleotide with all residues). Why is this? The authors are motivated by the GOR method for analyzing secondary structural propensities, but it is not clear that the same formalism is applicable here – namely, in the GOR method one analyzes the linkage between an object (amino acid) and its property, while here one analyzes the propensity of two objects (amino acid and nucleotide) to co-occur in the same context (i.e. contact). This is related to the asymmetry discussed in point 1.

4. The rather extensive literature on contact-based statistical potentials for nucleic-acid/protein interactions should be adequately cited and discussed (see, for example Donald et al. Nucleic Acids Res., 2007, 35, 1039–1047. or Tuszynska et al. BMC Bioinformatics, 2011, 12, 348 and other).

5. The authors refer to their randomized test set as a negative test set (p. 7). As there is no guarantee that many members of this set are not actual binders – the naming should be changed to something like “background set” or “randomized set”, but certainly not “negative set”. More critically, randomization was only done on the side of the RNA sequences (change of 1 nucleotide in the sequence) and not on the side of RRM sequences – this relates to the asymmetry of the whole approach as discussed above and must be properly defended.

6. Defining clusters as all complexes that have a certain similarity score with at least 25% of complexes in the cluster is quite low as a cutoff (p. 5). Of course, if one increases the cutoff, one risks not having sufficient samples for adequate statistics. The authors should defend the choice of their cutoff by providing quantitative evidence that it does not overly impact the scores i.e. the qualitative features of their method.

7. For the validation of their scoring method the authors analyze two experimentally studied examples, while extensive data on RRM binding motifs obtained by different experimental methods exists and is not used. See for example the RNAcompete results (Ray, D., Kazan, H., Cook, K. et al, A compendium of RNA-binding motifs for decoding gene regulation, Nature, 499, 172–177 (2013)) or the Attract database (PMID: 27055826). The authors should validate their results on an as extensive a set of experimental data as possible.

Minor comments

1. On p. 9, the authors state that “The unbiased number of observed contacts in the training set that is used to calculate the scores is also shown in the preference matrices (Figure 8 B,C), below each of the scores”. However, these numbers are not integers, so it is unclear what they actually refer to.  

2. Numbers in Figure 2 are not fully consistent with the text (1263 instead of 1259, 20 instead of 19; p. 3 and p. 4).

3. It is stated that the "alignment for the RRM-RNA structures" is available in Dataset S6 (p. 4). However, Dataset S6 contains the RRM-RNA similarity matrix.

4. In the caption of Table 1 (p. 28), the authors state that “The symbols reflect the score change after the E87N mutation”, however there are no symbols.

Reviewer #2: The paper describes construction of statistical based potential for scoring rrm domain interactions with specific RNA sequence. They use structural model of binding to identify contacting residues and then create sequence based interaction statistics. The authors acknowledge that the method is limited to already known binding modes of rna to RRM domains or very close to those. The authors validate the approach on leave out training set as well as few novel cases. The problem is still open - but there is a progress. I think the paper can be published. Would be interesting to compare the approach with AF like approach for modeling protein RNA interactions (see the link below). Also can protein rna models from this approach can be used to create alignment? https://www.biorxiv.org/content/10.1101/2022.09.09.507333v1.full.pdf

Reviewer #3: The manuscript by Roca-Martinez RNA-recognition motifs discusses a

scoring method to estimate binding between an RRM and a single

stranded RNA, and the method aims to predict RRM binding RNA sequence

motifs based on RRM protein sequence. The authors adopt a simpler

statistical approach over deep learning method employed in several

existing methods for better interpretability. Interesting results on

discriminatin of high affinity RNAs with the UAG core motif from lower

affinity RNAs are reported.

While the reported method serves a useful purpose towards the overall

task of solving the problem of deciphering RNA recognition code of

RRMs, there are a number of significant issues:

1. The score described by Equation 2 is not explained and the physical

unerpinning cannot be found. The two log terms summed are the same as

product of two ratio. But it is not clear what does it mean, and why

does this make sense? Does it model some eperical binding?

Conservation? Not clear. Furthermore, why the denominator in the

first term comes to be f_n - N_i,R_J ? This is not understandable.

2. There are numerous places where the method development depends on

visual inspection. This raises serious issure of reproducibility.

3. The model appears to be rather restrictive and works only for cluster 0 and no binding mode change can occur.

4. The negative test should be strenthened and should include other entries

if possible.

Other issues:

1. p.4. "the number of unique postions between both nucleotides" It is not clear if the authors meant positions only in A, only in B, or both?

2. Will the results sensitive to the specific threshold of 5A?

Minor issues

1. Figures seems to be jumping around in order, and it makes it difficult to go back and forth.

**Have the authors made all data and (if applicable) computational code underlying the findings in their manuscript fully available?**

Reviewer #1: Yes

Reviewer #2: Yes

Reviewer #3: Yes

PLOS authors have the option to publish the peer review history of their article (what does this mean?). If published, this will include your full peer review and any attached files.

Reviewer #1: No

Reviewer #2: No

Reviewer #3: No
---

## [Decision Letter · Decision Letter 1]

7 Jan 2023

Dear PhD Vranken,

We are pleased to inform you that your manuscript 'Deciphering the RRM-RNA recognition code: A computational analysis' has been provisionally accepted for publication in PLOS Computational Biology.

Best regards,

Shi-Jie Chen

Academic Editor

PLOS Computational Biology

Nir Ben-Tal

Section Editor

PLOS Computational Biology

Reviewer's Responses to Questions

**Comments to the Authors:**

Reviewer #1: The authors have significantly revised the manuscript and have adequately addressed all of my concerns from the first round.

Reviewer #3: The authors have made changes to improve the manuscript. However, a number of important issues have not been addressed adequately.

I find it still difficult to understand the origin of Eqn 2 and the physical basis remains unclear. Furthermore, while I appreciate why the authors are are using visual inspection for verifications, the issue of reproducibility largely remain.

**Have the authors made all data and (if applicable) computational code underlying the findings in their manuscript fully available?**

Reviewer #1: Yes

Reviewer #3: Yes

PLOS authors have the option to publish the peer review history of their article (what does this mean?). If published, this will include your full peer review and any attached files.

Reviewer #1: No

Reviewer #3: No

---

## [Editor Report · Acceptance letter]

17 Jan 2023

PCOMPBIOL-D-22-01066R1 

Deciphering the RRM-RNA recognition code: A computational analysis

Dear Dr Vranken,

I am pleased to inform you that your manuscript has been formally accepted for publication in PLOS Computational Biology. Your manuscript is now with our production department and you will be notified of the publication date in due course.

With kind regards,

Zsofia Freund
